# P-Renewal Project: A Reflexive Contribution to the Evolution of Energy Performance Standards for the Renovation of Historic Buildings

Sophie Trachte [1,*] and Dorothée Stiernon [2]

1   Faculté d'Architecture, Unité de Recherche Art, Archéologie & Patrimoine (AAP),
    Université de Liège (ULiège), B-4020 Liège, Belgium
2   Architecture et Climat, Louvain Research Institute for Landscape, Architecture, Built Environment (LAB),
    Université Catholique de Louvain (UCLouvain), B-1348 Louvain la Neuve, Belgium;
    dorothee.stiernon@uclouvain.be
*   Correspondence: sophie.trachte@uliege.be

**Abstract:** To meet European carbon neutrality targets and comply with building energy performance regulations, renovating historic buildings is considered one of the most challenging tasks for the construction sector. On one hand, commonly employed renovation solutions are often more difficult to implement on these structures. On the other hand, renovation work must be carried out while preserving their heritage value and integrity. The European standard EN 16883 on conservation and energy renovation performance of cultural heritage was developed in 2017 with the aim of facilitating energy performance improvements in historic buildings while respecting their cultural significance. In pursuit of the same objective, the "P-Renewal" project focuses on the energy retrofit of pre-war Walloon housing with heritage value, providing valuable support through a reflective process and decision-making tools. These tools enable the selection of renovation strategies that effectively combine the preservation of heritage value with improvements in internal comfort, energy efficiency, and environmental performance. This study compares the reflective process of the research project with the guidelines of the standard EN 16883 and discusses the transferability of this contribution to other European contexts. This will significantly contribute to the evolution of energy performance standards for the renovation of historic buildings.

**Keywords:** historic residential buildings; traditional residential buildings; energy renovation; standard EN 16883; reflexive process; multi-criteria analysis; decision-making tool

## 1. Introduction

This study first introduces the reflexive process offered by the P-Renewal research project, focusing on the energy renovation of pre-war Walloon heritage buildings, with the objective of effectively managing adequate energy retrofit interventions for old residential buildings with cultural or historical significance. The study then compares this approach with the guidelines outlined in the standard EN 16883 [1], titled "Conservation of cultural heritage—Guidelines for improving the energy performance of historic buildings".

The P-Renewal reflexive process was developed based on five case studies in Wallonia, the French-speaking region of Belgium. It adopts a bottom-up, inclusive, and multi-criteria approach, setting it apart in its originality. This distinctive approach involves considering energy performance improvement, indoor thermal comfort, and heritage value in a complementary manner. The aim is to achieve greater sustainability and reinforce the implementation of renovation strategies tailored to old buildings. In contrast to the standard EN 16883, the research project proposes a more structured and detailed reflexive renovation process, introducing new methodological steps. It offers robust and valuable support by providing advanced scientific and technical information on traditional buildings, such as existing energy performance, building systems, materials and services, heritage specificities, and common pathologies. Furthermore, it includes comprehensive details and illustrations

in an inventory of energy improvement solutions and provides building owners and professionals with decision-making tools, such as documentation templates, cross-tabulations, and decision trees. These diverse inputs can be regarded as a significant step forward, contributing real value towards the evolution of energy performance standards for the renovation of historic buildings.

A global shift toward a more sustainable and carbon-neutral economic system is currently underway. Within this transition, the construction sector plays a pivotal role in promoting the more efficient utilization of energy sources. Indeed, building stocks are notably energy-intensive, representing almost 40% of total final energy consumption [2], and constituting a significant contributor to $CO_2$ emissions in Europe. If the construction sector becomes more involved in the energy renovation of its building stock, it can actively contribute to the European Union's goals of reducing energy consumption and managing resources in a circular and sustainable way. Given that building materials contribute to 20–25% of the life-cycle carbon emissions in the current EU housing stock, adopting ambitious circular energy renovation strategies could save significant amounts of greenhouse gas emissions [2,3].

Since 2018, the European regulation on building energy efficiency [4] encourages EU member states to make their national existing building stock highly energy-efficient before 2050 by establishing long-term renovation strategies[1], inviting them to carefully screen their building stocks, organize renovation cycles based on a group of buildings rather than to individual properties and intensify the annual rate of renovation, currently around 1% to reach 2.5 or 3%.

The existing building stock represents a large number of dwellings[2], 82% in Wallonia and 64%, respectively, of the building stock in Europe [2], with a predominance of single-family houses [2,5].

This building stock is also considered energy-intensive [2]. Indeed, by analyzing datasets, nearly half of Walloon residential buildings [6] and more than 40% withing the existing European stock [2] are built before the 1960s with low insulation levels and old, inefficient technical systems (heating and ventilation). This low performance implies that a significant number of these buildings must undergo energy performance improvements by 2050. Typically, up to 30% energy savings can be expected by applying one to three inexpensive and easy-to-implement measures, such as a new boiler or loft insulation. However, achieving near-zero levels of energy consumption and carbon emissions requires the complete insulation and airtightness of the envelope's walls, the integration of a ventilation system, and the replacement of all elements influencing energy use (windows, heating system, etc.), along with the installation of renewable energy technologies [2]. However, this Walloon and European dwelling stock is also made up of historic and traditional buildings as defined by Webb[3], for which insulation measures and commonly used technical systems are often more difficult to apply. Webb defends the idea that historic and traditional buildings account for a sizeable proportion of it [7]. According to Troi[4], 14,3% of the existing buildings in EU-27 were built before 1919, and 12.1% were built between 1919 and 1945 [8], while for Belgium, the corresponding values are 11.5% and 12.7%, respectively. Nevertheless, Wallonia has a lot of older housing, with almost a quarter of all dwellings built before 1921 [9].

### 1.1. State-of-the-Art

The energy and sustainable renovation of historic and traditional buildings with heritage specificities, protected by conservation measures or not, has thus become a major challenge for Wallonia and many other European regions.

This is why, in the last decade, scientists have shown a growing interest in the energy renovation of historical buildings, combining thermal comfort, heritage value, and energy efficiency [10–12]. This interest has resulted in numerous research projects, scientific articles, and conferences focusing on traditional and historic building stock, specifically on the pre-war-dwelling stock and its energy efficiency improvement.

The first series of research projects has focused on the typological analysis of the existing built stock, historical, traditional, and/or more contemporary, aiming to estimate the energy performances of this building stock and propose, through exemplary renovations or case studies, energy improvement measures. These include Intelligent Energy Europe (IEE) projects TABULA [13]—EPISCOPE [14] at the European level and, at the Belgian level, research projects Low Energy Housing Retrofit (LEHR) [15], B³RetroTool [16], cost-optimum (CO-ZEB) studies [17], and studies of Centre d'Etude, de Recherche et d'Action en Architecture (CERAA) [18], C. Kints [19], and Atelier parisien d'urbanisme (APUR) [20]. This series of studies and research has demonstrated that energy renovation of traditional and historic buildings has a major role to play in the transition to a sustainable and neutral-carbon energy system and that historic and traditional buildings could easily accept several energy improvements that can significantly reduce the energy demand of the European building stock. According to an estimate of Troi, the retrofit of European dwellings built before 1945 could have saved an annual equivalent to 3.6 percent of the total EU-27 $CO_2$ emissions in 1990 [8]. Those studies also highlighted the diversity as well as the architectural and landscape quality of traditional and historic building stock [21,22].

A second series of research, such as the BAPE [23] and Eco-Rénovation patrimoniale Formation et INnovation (ERFIN) projects [24], has demonstrated the need to develop a better knowledge of historic and traditional buildings and mainly buildings composed of massive walls in natural stone and brick [25,26]. Within this objective, various research projects, such as BATAN [27], HYGROBA [28], and HUMIBAtex [29], also focused on the study of the physical phenomena that characterize the thermal behavior of traditional buildings, on the analysis of hygrothermal transfers in traditional buildings as well as on the development of calculation models to assess their thermal behavior [30,31]. At least several research projects, such as [32], have set out to develop multi-disciplinary methodological approaches for investigating buildings to document and characterize existing traditional buildings deeply. This information is crucial to performing energy diagnostics [21], developing energy models, and, based on these models, judging the hygrothermal risks associated with a particular renovation measure [33]. Studies on heritage buildings have encouraged the development of non-invasive technologies for such documentation and analyses [34–39].

This second series has demonstrated that historic and traditional buildings have a specific thermal and hygrothermal behavior characterized by the high thermal inertia of walls, specific ventilation/infiltration patterns, and the presence of specific internal spaces serving as a thermal buffer between heated spaces and the outside. Very dependent on the local environment [22], they are built with "moisture-sensitive" materials (raw earth, terracotta, wood, etc.), which are also the seat of complex coupled HAM (heat, air, and moisture) transfers, not simple to model. At least the presence of building systems is often limited or outdated, especially the ventilation system.

Finally, a last series of research projects have mainly focused on the proposition and evaluation of solutions to improve the energy efficiency of traditional and historic residential buildings, both on the envelope [40–45] on technical systems such as heating and ventilation systems [12,46,47] and on renewable energy supply [48]. These also include works performed in SHC IEA tasks [49,50]. Most of these research projects relied on exemplary renovations and/or study cases to propose methods for selecting improvement solutions and/or to provide some innovative technologies and materials [51–53]. Some projects have focused more on energy improvement solutions for built heritage in urban areas [54] and historic districts [55]. Through case studies, those projects have proposed both energy improvement solutions applicable to most of the European-built heritage in urban areas [56] and a methodology with criteria for selecting and prioritizing energy efficiency interventions in historic districts [57]. More recently, research works performed in the SHC IEA task 59 [58] and its IEA EBC Annex 76 [59] have developed a database of exemplary buildings for energy renovation on old, traditional, and historic buildings presenting in detail and for each building, with conservation compatible energy renovation solutions having been integrated [60–63]. These

projects also provided a more holistic approach to the energy renovation of historic buildings by working both on the documentation and the evaluation of technical solutions (including the contribution of renewable energies) and/or renovation strategies, which allow the preservation of the historic and aesthetic values while increasing comfort, lowering energy cost, and minimizing environmental impact [64].

All these studies have led to various conclusions. First, each historical and/or traditional building must be considered as a particular case where energy measures must be proposed and assessed based on sufficient documentation of the constructive, architectural, and spatial specificities as well as on a detailed diagnosis specifying the potential pathologies and hygrothermal behavior of the building [65]. In addition, any intervention must be governed by a detailed understanding of the building's heritage significance, including its heritage values and character-defining elements. This understanding enables us to make appropriate decisions when energy renovation measures are proposed and/or implemented and guarantees the building's conservation and preservation as well as the respect of its historic values. Secondly, energy improvement measures can be undertaken both on the building envelope and on technical systems. Several measures can complement each other and thus be combined in an energy renovation strategy, but the compatibility among them must be carefully considered before implementation and the hierarchical order in which they are to be undertaken [66]. Finally, the various aspects of energy performance and heritage conservation must be considered in a broader context of sustainable renovation to ensure its long-term use and low overall environmental, economic, social, and cultural impact [67].

Suppose all these projects have enabled a considerable advance in scientific knowledge in the field of energy renovation of traditional and historic buildings. In that case, this knowledge has not yet been acquired and/or applied in the practices of energy renovation professional actors despite the development of specific decision-making tools and guidelines all around Europe [60]. Indeed, the reality experienced by professionals is more complex because it incorporates further considerations about the conservation of old buildings and architectures, including timing, technical, economic, and social considerations [60]. Moreover, in any case, renovation solutions are not directly transferable to other cases. The requirements differ from case to case regarding various factors. Although buildings within the same typology exhibit numerous similarities, each presents distinct construction features, particularly evident in the nature, sections, or thicknesses of materials utilized. For example, the roof structure type varies significantly based on the building's size (trusses or purlins and rafters, wood sections, etc.). The state of preservation, notably issues related to humidity-induced pathologies, differs across situations, significantly impacting the selection of energy improvement measures. Geographical location and climate play a central role in determining whether the improvement measures focus more on comfort in winter or summer within a bioclimatic design approach. Additionally, the occupant's profile, whether an owner or tenant, and their lifestyle significantly influence the choice of improvement measures [68]. These factors are often overlooked or inadequately addressed in scientific studies and renovation guides. Their consideration is frequently compartmentalized, missing the holistic perspective necessary for comprehensive building enhancements. This has resulted in a gap between practice and theory in balancing preservation, use, and energy efficiency in historic buildings.

To fill the gap between theory and practice, the European Standard "Conservation of cultural heritage, Guidelines for improving the energy performance of historic buildings" [1] was developed and published in 2017. It is meant to be used by building owners, authorities, and professionals involved in the energy renovation of historic buildings and provides a decision-making framework and guidelines for sustainably improving the energy performance of historic buildings while respecting their heritage significance.

A first study carried out in the framework of the IEA task 59 project has analyzed and evaluated the practical usability of the standard to make suggestions for enhancing the European guidelines for improving the energy performance of historic buildings [69]. The evaluation was carried out in 13 countries through professionals' networks, and study

cases presenting different scales, uses, and complexity were studied at different stages of the process planning. The evaluation has highlighted the main strengths of the current standard, such as the generic decision-making framework, the interdisciplinary and case-by-case approach, and the balanced focus on both heritage and energy aspects. It has also outlined various weaknesses, such as the difficulty of interpreting the guidelines, i.e., translating abstract and generic steps into actions, the lack of examples of how to perform tasks, and the difficulty of integrating the logic of the process planning into the working practices, especially for small renovation projects [69,70]. A second study has described how the standard can be applied in practice to heritage value assessment, building surveys, and holistic assessments of energy efficiency measures [70]. Accompanied by exemplary study cases, this study has illustrated how the different stages of the planning process can be carried out in practice. This study has also highlighted the value of including in the reflexive processexemplary study cases and decision-making tools for assessing both the impact of proposed renovation measures or strategies on the conservation of heritage value and the overall environmental impact.

However, the main conclusion of those two studies was that professionals and practitioners rarely use the standard EN16883, while researchers have shown real interest in the field of energy efficiency in historic buildings. Practitioners and professionals do not directly identify the benefits or the positive impact brought by using it [69]. They also point to the time and resources required by the procedure, especially if it is considered at the very beginning of the process planning. The two studies have suggested improving its usability with decision-making tools to support all stages of the reflexive process, including the phase of building survey, as well as with several types of examples, including examples of how the steps of the procedure can be carried out or examples of energy efficiency measures, examples.

### 1.2. Aim of the P-Renewal Project

Funded by the Energy Department of the Walloon Public Service as a part of the "Energie—IEA Program" call for research projects and launched the same year as the standard EN 16883 on the conservation and energy renovation performance of cultural heritage, the P-Renewal research project [71] was conducted in collaboration with the Unit Building Performances & Renovation of the Belgian Building Research Institute (BBRI, actually Buildwise). The project shares the same objectives as the standard EN16883, namely: (1) facilitate and support the planning process during energy renovation of historical and traditional residential buildings built before 1919 (listed or not), and (2) provide decision-making tools to help owners and design professionals to characterize the qualities and needs of historic residential buildings and adequately select renovation measures to improve their internal comfort and overall environmental performance while preserving their heritage value. A third objective completes the first two: assessing whether the energy requirements of the Walloon Region can be met for this building type and at what cost [72]. If this project aligns with the objectives and the guidelines of the standard EN 16883, it also introduces specific innovations to enhance the usability of the planning process. Additionally, it provides technical data on pre-war traditional residential buildings, drawing from the analysis of five case studies. Most importantly, for each decision step, it offers one or more tools to facilitate the user's decision-making process.

The study is structured as follows: Section 2, "Methodology", presents the planning process proposed by the standard as well as the methodological approach of the P-Renewal project. Section 3, "Results and Discussion", outlines the various data, information, and decision-making tools collected and developed within the research project and discusses the contribution of these tools in enhancing the usability of the standard EN16883. Section 4, "Conclusions and Perspectives", outlines the conclusions of the work and introduces possible further developments.

## 2. Methodology

### 2.1. Retrofit Planning Process Proposed by the Standard EN 16883

The standard EN 16883 on the conservation and energy renovation performance of cultural heritage was developed in 2017 with the aim of facilitating the decision-making procedure during the energy renovation of historic buildings by integrating measures for energy performance improvements and reduction in greenhouse gas emissions with adequate conservation of the building. It can be applicable to historic buildings of all types and ages, listed or not. It proposes a systematic procedure, presented in Figure 1, to support, in each specific case, the selection of appropriate measures in the planning stage of the renovation while respecting a sustainable balance between the use of the building, its energy performance, and its conservation. This procedure is built on a multidisciplinary and integrated approach considering the collaboration of owners and users of the building, as well as on a thorough knowledge of the building's structural, constructive, and technical specificities, as well as heritage values and character-defining elements. It also proposes a stage of renovation measure evaluation combining the assessment of technical compatibility, economic, heritage, outdoor environment, and practical implementation aspects with energy performance assessment.

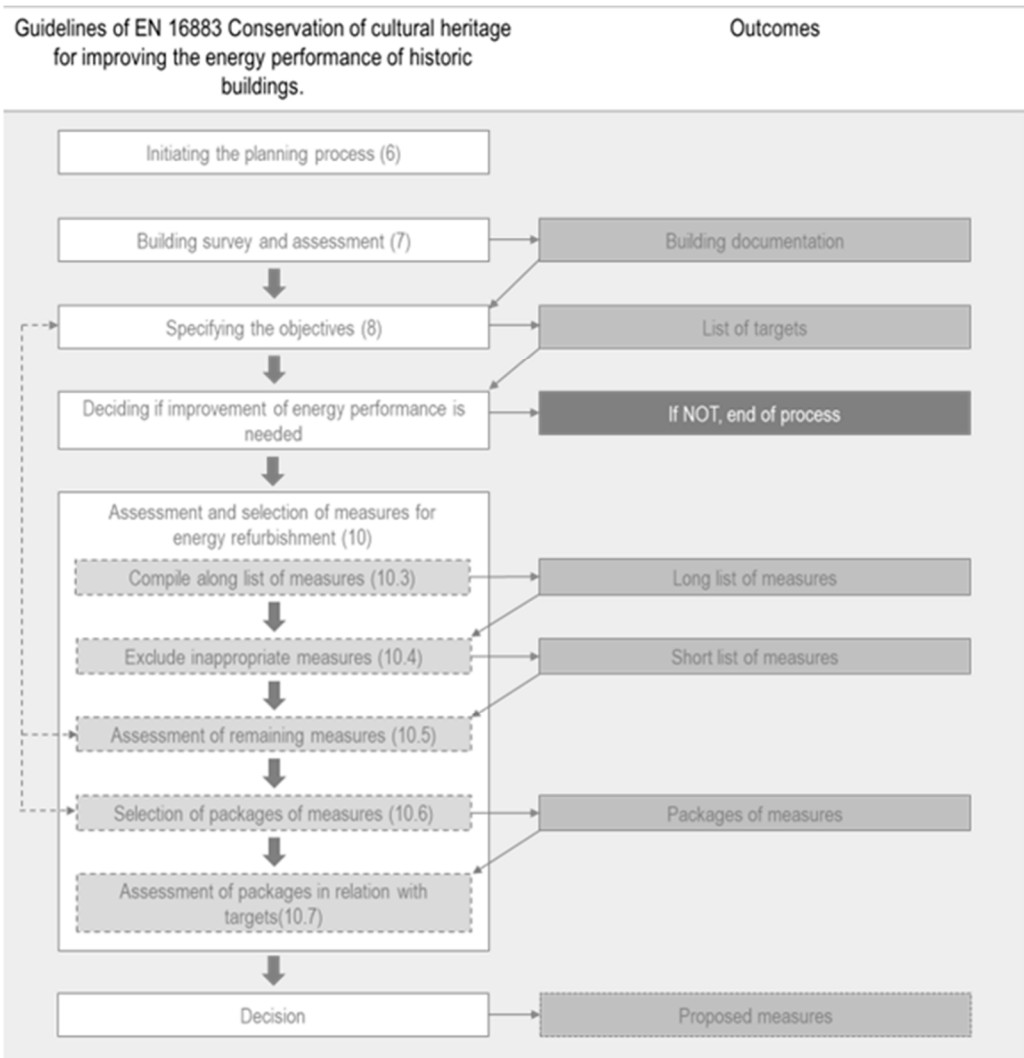

**Figure 1.** Renovation procedure and outcomes proposed by the European standard EN 16883.

The standard is structured into eleven chapters. The first five chapters define the scope, the normative references, and specific terms related to the building's renovation and refurbishment. They also provide a general consideration of the building's energy improvement and an overview of the proposed procedure. Then, as shown in Figure 1, chapters 6 to 11 present the various steps of the decision-making procedure and the various outcomes, from initiating the planning process to the final decision. The first steps (6 to 8) focus on identifying the need for energy improvement, while step 10 provides a set of guidelines to adequately select a set of energy renovation measures and assess them according to the target objectives. The standard is also accompanied by documents providing templates to facilitate the documentation of the building (Annex A) and the qualitative assessment of the selected measures (Annex B).

*2.2. Methodological Approach Proposed by the P-Renewal Project*
Bottom-up, Integrated, and Multi-disciplinary Methodological Approach

The P-Renewal research project is consistent with the objective of the standard EN16883 by providing owners and professionals with a holistic, reflexive planning process supported by decision-making tools to adequately address the challenges of energy performance improvement of pre-war traditional and historic residential buildings. The interest of the reflexive process is both its bottom-up (case-by-case) and integrated approach, as well as its multi-disciplinary character since it combines the improvement of the building's energy performance with the diagnosis and remediation of pathologies, the occupancy potential of the building, the improvement of indoor comfort and the global environmental performance.

- Bottom-up approach

The reflexive process has been developed based on five building study cases sufficiently representative of the main types of pre-war Walloon dwellings. Those five study cases have enabled both the implementation of a thorough survey and diagnosis stage focusing on the four dimensions (energy, comfort, heritage value, and environmental performance) of the project and the evaluation of the tools developed to support the decision-making process since each stage of the project has been confronted with those five buildings. Data collected on study cases have also allowed the development of dynamic simulation models to evaluate energy improvement solutions and strategies. This bottom-up approach has thus made it possible to consider the specificities and characteristics of different pre-war building types and to fit more into a case-by-case decision-making process.

- Integrated approach

The target audience of the P-Renewal project was owners and professionals in the field of energy efficiency in historic residential buildings. Professionals have been fully integrated into the research project due to the sponsorship of the professional federation Embuild [73] and the Walloon Heritage Agency [74] and the collaboration of a "User Group" made up of architects, renovation companies, energy efficiency experts, as well as heritage refurbishment experts.

The User Group was invited semi-annually to participate in meetings for presentations and discussions regarding research progress, achieved results, and key considerations for future steps. Specific aspects, including renovation objectives, the inventory of energy improvement solutions (envelope, technical systems, and renewable energy), and the selection of strategies, were thoroughly reviewed and validated by the User Group. Some experts within the group actively contributed to the development of specific deliverables by providing their technical expertise.

Considering the activities of the User Group members, the reality experienced on building sites, and their needs, those meetings and discussions were considered valuable support in developing the research and deliverables. They significantly enhanced the robustness and usability of the reflexive process, facilitating a swift transition from theoretical concepts to practical applications in the field.

- Multi-disciplinary approach

The reflexive process and the selection of energy improvement solutions and renovation strategies were based on a multi-disciplinary approach combining the energy performance improvement with the potential occupancy of the building, the comfort improvement, the conservation of heritage value, the financial cost, and the overall environmental impact. These various criteria were not hierarchically prioritized or weighted. The objective of the research project was to evaluate the impact of enhancing energy performance and comfort on the other dimensions of the building, whether positively or negatively. This approach makes it possible to integrate energy performance improvement into a broader vision of sustainable building renovation, which integrates energy performance and heritage value aspects considering economic, social, and environmental criteria.

### 2.3. Complementary and Iterative Methodological Steps, Following Guidelines of the Standard EN 16883

To offer a more comprehensive and holistic vision in addressing the challenges of the improvement of the energy performance of pre-war Walloon residential buildings, the P-Renewal has been conducted into various complementary and iterative methodological steps to develop and structure a renovation reflexive planning process adapted to pre-war historic and traditional buildings and their specificities. The research project has also provided a series of technical data and decision-making tools to support the reflexive process.

Those various methodological steps are illustrated in Figure 2 and presented below by highlighting the objective pursued, the method used, and the data collected for each of them. The whole reflexive process and the various tools developed will then be presented and discussed in Section 3, "Results and Discussion".

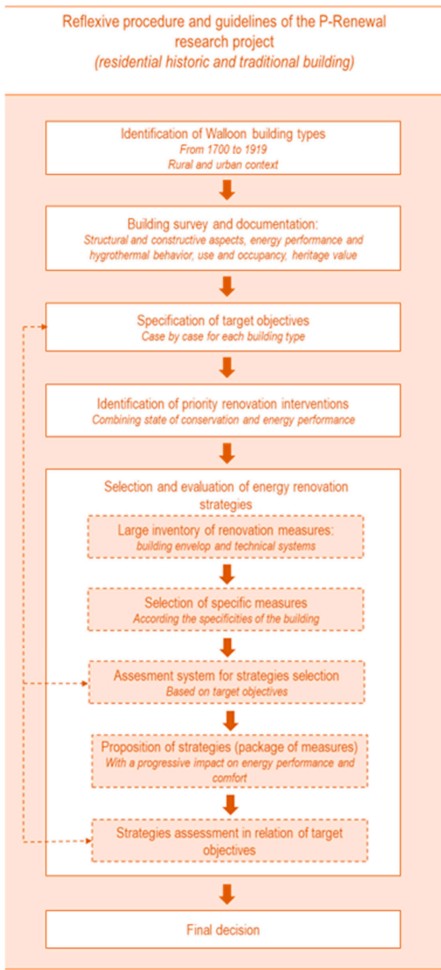

**Figure 2.** Renovation reflexive procedure proposed by P-Renewal project.

2.3.1. Typological Analysis of the Walloon Pre-War Building Stock

A typological analysis of dwellings built before 1914 was performed on the Walloon building stock with the objective of better understanding the constructive specificities of the built stock existing in Wallonia [75]. In fact, the renovation planning process must start by acquiring data and knowledge on the building to be renovated to guarantee the adequacy and quality of the selected energy improvement measures as well as the durability of the building. It is first essential to collect a certain amount of information to develop a sufficiently detailed picture of the building before making assumptions about its energy performance and/or collecting more information about its state of conservation and heritage specificities. This stage of data acquisition can be established quickly and easily if the building type to which the studied building refers can be clearly identified.

This analysis has been carried out based mostly on the literature [16,76–81] and according to several indicators such as the context, positioning in relation to the public space, general dimensions, organization of internal spaces, construction, and material specificities. The study allowed the identification and description of the main types of historical and traditional housing in Wallonia, both in rural and urban contexts. It was completed by a numerical appraisal and a geographical distribution carried out using the database of the Regional 2016 Property Register [82].

2.3.2. Selection of Study Cases

With the objective to develop a case-by-case decision-making process, five buildings considered sufficiently representative of the main pre-war residential building types in Wallonia were selected as study cases, with the support of the User Group, the Walloon Heritage Agency, and the professional federation Embuild.

The selection, shown in Figure 3, considered both the two main construction periods in Wallonia and the geographical distribution to cover a significant part of the Walloon Region, especially the area "Sambre and Meuse", which is the most densely built. It also considered the authenticity and integrity of the study cases, most of which have undergone little or no renovation work since their construction.

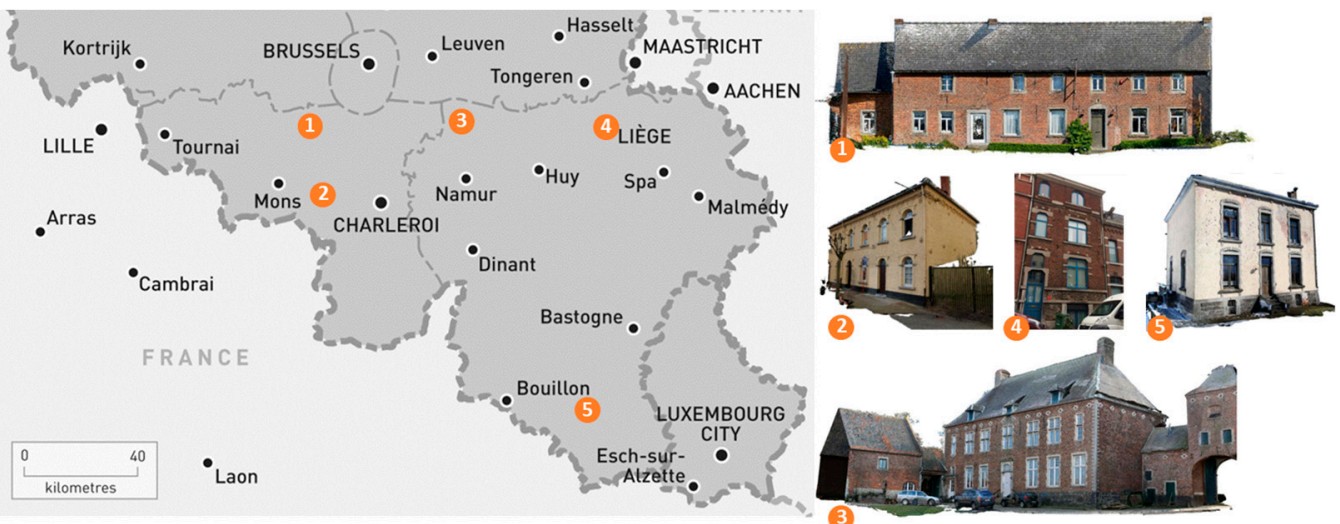

**Figure 3.** Geographical situation of the five study cases used in the P-Renewal project: 1. A multi-cellular farm, dating from 1789, in Enghien. 2. A worker house in Bois-du-Luc, a mining site inscribed on the UNESCO (United Nations Educational, Scientific, and Cultural Organization) World Heritage List, dating to the late 19th century. 3. A listed farm with a courtyard built in 1754, in Jodoigne. 4. A typical "bourgeois terraced house" in Liège from the early 20th century. 5. A "villa", on a listed area, built in 1617, in Léglise.

The number of study cases was intentionally limited to study them in detail, knowing that on-site analyses and building monitoring may be time- and resource-consuming.

### 2.3.3. Documentation of Study Cases

Thorough surveys and diagnostics were then carried out on the five study cases in deep collaboration with Buildwise. The objective of those surveys was to properly describe their characteristics, evaluate their strengths and weaknesses in terms of energy performance, building quality, and internal comfort, as well as to better understand the hygrothermal behavior of its walls and components and identify potential pathologies, heritage specificities, and habitability potential [65].

As illustrated in Table 1, surveys have included general information about each study case, a comprehensive technical and physical analysis documenting its structural type, components, state of conservation, and energy performance, as well as a description of its heritage significance and character-defining elements, and a study of its occupancy/use potential.

**Table 1.** Type of data collected on the five study cases during the diagnostic step.

| **Preliminary Surveys** | |
| --- | --- |
| Building site and local environment | Description of the local context and external spaces. General description of the building. |
| Building technologies and materials | Description of constructive system, material components (nature and thickness), and the state of conservation of roof and roof structure, attic, façade walls, windows (frame and glazing), doors, slab and floors, and internal walls. |
| Spatial organization | Description of spatial organization and natural lighting of the ground floor, first floor, second floor, attic, and ceiling (if exits). |
| Building services and technical systems | Description and state of the conservation of the sewerage network and stormwater collection, heating system, ventilation system, electricity network, and lighting. |
| Statements of energy consumption | Analysis of energy bills, in agreement with the occupant. |
| Discussion with occupants | Discussion about the perceived advantages and weaknesses in terms of comfort and the use of the building. |
| Geometric documentation | Photogrammetry and 3D surveying techniques. |
| **In-depth analysis (energy efficiency and comfort)** | |
| Diagnostics | Thermography (with camera). Flatness (for façades and floors). Salts and moisture. Air tightness (with blower door test). |
| Energy efficiency and thermal comfort | U-value estimation. Monitoring (temperature and relative humidity). |
| **Complementary analysis** | |
| Heritage specificities | Description of building heritage specificities<br><br>- In general;<br>- External: façades walls, windows, doors, and roofs;<br>- Internal: finishes and decorations. |
| Occupancy potential | Development and proposal of various occupancy scenarios allowing the densification and/or expansion of certain spaces (by extension, annex, etc.) |

Each case study was investigated and described in detail, compiling as much technical data as possible. For this purpose, innovative 3D surveying techniques were extensively used to characterize the building's geometry and state of conservation, as shown in Figure 4. Combined with this, wireless sensor networks were deployed in every test building to monitor the conditions of the indoor air finely over a long period. More specific tests were performed to complement this monitoring campaign, including air tightness and thermographic measurements [65], and inhabitants were interviewed during the on-site visits.

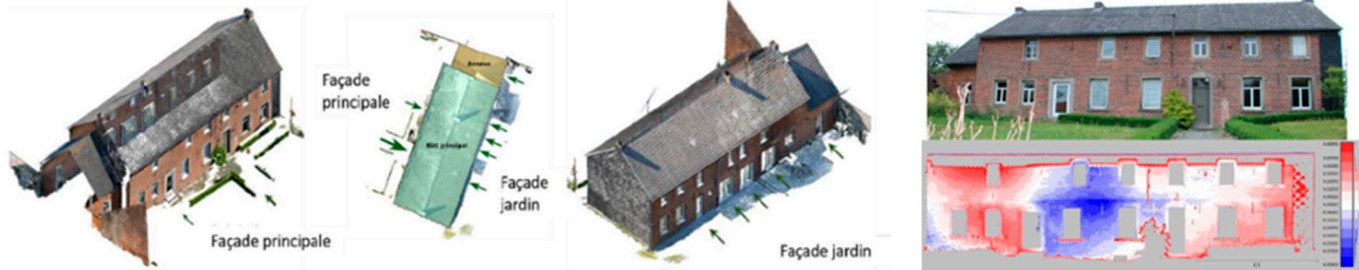

**Figure 4.** 3D surveying techniques allowed the collection of a large quantity of technical data. Source: Unit Building Performances & Renovation, Buildwise—S. Dubois [72,82].

Data collected on each wall of the study cases (materials, nature, and thickness) allow U-value estimation, as shown in Table 2.

**Table 2.** Estimation of the thermal loss coefficients (U-value) for the walls of each case study. Estimation conducted using default values outlined in Appendix of the standard NBN 62-002 [83].

| | Study Case 01 | Study Case 02 | Study Case 03 | Study Case 04 | Study Case 05 |
|---|---|---|---|---|---|
| **Roof** | $5.18 \text{ W/m}^2\text{K}$ | / | $5 \text{ W/m}^2\text{K}$ | / | $4.1 \text{ W/m}^2\text{K}$ |
| **Attic floor** | $0.63 \text{ W/m}^2\text{K}$ | $1.8 \text{ W/m}^2\text{K}$ | Between 2.00 and $0.74 \text{ W/m}^2\text{K}$ | $0.41 \text{ W/m}^2\text{K}$ | $1.44 \text{ W/m}^2\text{K}$ |
| **Façade wall** | Between 1.15 and $1.56 \text{ W/m}^2\text{K}$ | $1.42 \text{ W/m}^2\text{K}$ | Between 2.17 and $1.29 \text{ W/m}^2\text{K}$ | Between 2.33 and $1.25 \text{ W/m}^2\text{K}$ | Between 1.32 and $1.29 \text{ W/m}^2\text{K}$ |
| **Window** | Very old and old wood frame with simple or double glazing | Old PVC frame with double glazing | Old wood frame with simple glazing | Old wood frame with double glazing | Old wood frame with double glazing |
| **Cellar floor** | / | $2.6 \text{ W/m}^2\text{K}$ | Between 0.75 and $0.70 \text{ W/m}^2\text{K}$ | Between 2.03 and $1.83 \text{ W/m}^2\text{K}$ | Between 2.05 and $1.92 \text{ W/m}^2\text{K}$ |
| **Slab or floor basement** | Between 1.86 and $2.46 \text{ W/m}^2\text{K}$ | / | / | Between 3.48 and $2.77 \text{ W/m}^2\text{K}$ | Between 3.6 and $3.38 \text{ W/m}^2\text{K}$ |

Although a classification measure did not protect some of the study cases, the analysis of heritage significance was carried out according to four criteria (authenticity, integrity, rarity, and representativeness) and eleven indicators proposed by the Walloon Heritage Administration assessment method[5]. All these data have been analyzed on-site and collected in a report where a synthesized conclusion was presented in the form of a table containing a qualitative evaluation with "++", "+" and/or "-" for each heritage interest, depending on its level of importance. An example of qualitative evaluation is presented in Table 3.

**Table 3.** Qualitative evaluation of the architectural interest according to the four criteria of the Walloon Heritage Administration (SPW, 2020a)—study case "multi-cellular farm". NA is used when one of the criteria is not applicable.

| Architectural Interest | Walloon Heritage Administration's Criteria | | | |
|---|---|---|---|---|
| | Authenticity | Integrity | Rarity | Representativeness |
| Functional organization—outdoor | - | - | NA | + |
| Spatial organization—indoor | ++ | + | NA | ++ |
| Volume and size | ++ | ++ | NA | ++ |
| Constructive systems | ++ | + | NA | ++ |
| Main materials | + | + | NA | ++ |
| Façade composition | - | - | NA | + |
| Façade materials | + | + | NA | ++ |
| Roof materials | ++ | + | NA | + |
| Slab composition | - | - | NA | + |
| Floor composition | ++ | + | NA | ++ |
| Slab and floor materials | + | - | NA | + |

Furthermore, a reflexive study on the occupancy potential has been conducted with the objective of proposing several alternatives in terms of building use. Although the standard makes no reference to it, the question of occupancy and building densification must also be considered during the renovation planning process because most traditional residential buildings, currently used as dwellings, present a significant potential to evolve into different use types and/or to be densified [81,84,85]. This densification also positively impacts the global energy consumption/demand of historic buildings and can be considered a lever for European cities' energy and environmental transition.

The technical and physical data collected on the study cases provided the input parameters necessary for the development of dynamic simulation models (via DesignBuilder software v5) used for the selection and evaluation of renovation solutions and strategies as well as for the critical comparison with Walloon's EPB (Energy Performance of Building) software. They also provided thorough information about the strengths and weaknesses regarding each study case's comfort and energy performance, as well as heritage specificities that must be conserved and preserved. Those data supported the definition of renovation objectives, of the selection of the main interventions to be carried out, and the identification of potential renovation measures and strategies.

As an example, Table 4 presents, for each case study, three modeled heating scenarios proposed to approximate the annual heating demand (based on energy bills) and calibrate the simulation model. Heating scenario 01: all spaces, including cellars and areas not directly occupied by residents, are maintained at a temperature of 20 °C; heating scenario 02: Inhabited living spaces are maintained at 20 °C, walkways at 15 °C, and unoccupied areas such as cellars and attics are heated to 10 °C; and heating scenario 03: inhabited living spaces are heated to 21 °C, passageways and circulations are maintained at 15 °C, while unoccupied rooms such as cellars and attics remain unheated.

**Table 4.** Estimation of annual heating requirements by dynamic simulation.

| | Estimation of Annual Heating Demand Based on Energy Bills | Estimation of Annual Heating Demand—3 Heating Scenarios | | |
|---|---|---|---|---|
| | | Scenario 01 | Scenario 02 | Scenario 03 |
| **Study case 01** 223 m² | 71.48 kWh/m² | 109.59 kWh/m² | 57.49 kWh/m² | 103.40 kWh/m² |
| **Study case 02** 57 m² | 332.98 kWh/m² | 94.27 kWh/m² | 68.99 kWh/m² | 176.77 kWh/m² |
| **Study case 03** 228 m² | 186.38 kWh/m² | 124.85 kWh/m² | 80.12 kWh/m² | 78.27 kWh/m² |
| **Study case 04** 137 m² | 223.86 kWh/m² | 92.56 kWh/m² | 59.27 kWh/m² | 108.72 kWh/m² |
| **Study case 05** 440 m² | / | 126.03 kWh/m² | 103.89 kWh/m² | / |

2.3.4. Specification of Renovation Target Objectives, Following a Case-by-Case Approach

With the aim of encompassing energy performance enhancements within a broader framework of sustainable renovation, the P-Renewal research project has delineated multiple targeted renovation objectives. These objectives integrate considerations such as the state of conservation, internal comfort, energy performance improvement, and overall environmental performance. The significance of global environmental performance is particularly emphasized, given that traditional and historical buildings were typically constructed in harmony with their surroundings, utilizing available resources and adapting to climatic constraints [7,77,78].

Specific renovation objectives concerning comfort, energy, and environmental performance were proposed for each case study. Subsequently, these objectives were deliberated with the User Group, the Walloon Heritage Agency, and the professional federation Embuild.

Comfort objectives were determined based on criteria and indicators outlined in the Walloon Housing Code [86] and the Housing Quality Survey 2007 [87]. Energy improvement goals were set with the aim of achieving the "PEB-A" level of the Walloon Retrofitting Strategy [6], which corresponds to a primary energy consumption between 45 and 85 kWh/m$^2$. However, it is worth noting that this energy level is considered optimistic for old buildings with heritage value. Environmental objectives were formulated to enhance the functionality of each case study and improve the quality of life for occupants, all while minimizing the building's impact and its environmental footprint throughout its life cycle, including during renovation operations. Special attention has been paid to greenhouse gas emissions, the use of natural resources (including grey energy), and waste generation.

This exhaustive and in-depth discussion on the various target renovation objectives allowed us to define an innovative holistic renovation approach, probably more suitable for the energy renovation of old historic and traditional buildings [88].

2.3.5. Identification of Priority Renovation Interventions

After gaining approval for the target renovation objectives from all stakeholders, a comprehensive list of potential renovation interventions for both the building envelope and systems was outlined. These interventions were subsequently organized, classified, and prioritized with the goal of pinpointing those that would yield the most significant impact on both energy improvement and occupant comfort. To better determine the priority level of each intervention, the data gathered for the walls of each study case's envelope were analyzed concerning both their state of conservation and energy performance, specifically in terms of the thermal transmission coefficient (U-value). Initially, both sets of data were evaluated and assigned a ranking on a scale of 1 to 4, signifying the priority level for maintenance intervention and energy consumption, respectively, as outlined in Tables 5 and 6. For the U-value ranking, the first and second columns, "very and energy-intensive", present the values encountered in the case studies, and the last column, "energy efficiency", presents values slightly lower than those required by Walloon regulation but considered by experts as "energy efficient" and the third column "xx" presents values regularly found by the experts of the User Group and considered as moderately energy efficient.

Subsequently, these rankings were cross-referenced in a matrix, presented in Section 3, "Results and discussion", to identify the priority for energy improvement.

Furthermore, discussions with the User Group have underscored the prevalence of energy renovation projects lacking proper planning. These endeavors often occur urgently, without a thorough consideration of essential questions or the effective management of potential interactions between energy renovation measures and other aspects such as habitability, building quality, and environmental performance. This approach leads to prolonged work times on-site, elevated financial costs for the renovation, and, at times, necessitates dismantling recent renovation interventions.

**Table 5.** Table defining the envelope state of conservation and priority level of renovation or maintenance intervention.

| | Priority Level of Renovation's Intervention According to the State of Conservation | | | |
|---|---|---|---|---|
| | **Priority** | **Necessary** | **Possible** | **Nonpriority** |
| **Pitched roof, wood frame** | Wood frame and roof covering in poor condition | Wood frame in good condition but lack of watertightness | Wood frame and others roof layers with normal ageing state, without degradation | New or renovated for less than 5 years |
| **Flat roof, wood frame** | Wood frame and water-tightness in poor condition | Wood frame in good condition but watertightness to be replaced | Wood frame and others roof layers with normal ageing state, without degradation | New or renovated for less than 5 years |
| **Front façade** | Façade with a lot of degradations | Façades frame with small and/or sporadic degradations | Façade with normal aging state, without degradation | New or renovated for less than 5 years |
| **Rear and other façades** | Façades with a lot of degradations | Façades frame with small and/or sporadic degradations | Façades with normal aging state, without degradation | New or renovated for less than 5 years |
| **Window (frame and glazing)** | Single glazing, window frame in poor condition, lack of airtightness | Single or old double-glazing window frame with sporadic degradations | Double glazing, window frame with normal aging state, without degradation | New or renovated for less than 5 years |
| **Slab** | Slab in poor condition, including finishing | Slab in good condition, finishing with significant degradation | Slab with normal aging state, without degradation | New or renovated for less than 5 years |

**Table 6.** Table defining the energy performance of wall's envelope.

| | Energy Performance of Wall's Envelope | | | |
|---|---|---|---|---|
| | **Very Energy Consuming** | **Energy Consuming** | **Low Energy Consuming** | **Energy Efficient** |
| **Pitched roof, wood frame and/or attic floor** | No insulation layer $U > 1.5$ W/m$^2 \cdot$K | Thin insulation layer $1.5 < U < 1$ W/m$^2 \cdot$K | Medium insulation layer $1 < U < 0.4$ W/m$^2 \cdot$K | Thick insulation layer $U < 0.4$ W/m$^2 \cdot$K |
| **Flat roof, wood frame** | No insulation layer $U > 1.5$ W/m$^2 \cdot$K | Thin insulation layer $1.5 < U < 1$ W/m$^2 \cdot$K | Medium insulation layer $1 < U < 0.4$ W/m$^2 \cdot$K | Thick insulation layer $U < 0.4$ W/m$^2 \cdot$K |
| **Front façade** | No insulation layer $U > 3$ W/m$^2 \cdot$K | Thin insulation layer $3 < U < 1.8$ W/m$^2 \cdot$K | Medium insulation layer $1.8 < U < 0.5$ W/m$^2 \cdot$K | Thick insulation layer $U < 0.5$ W/m$^2 \cdot$K |
| **Rear and other façades** | No insulation layer $U > 3$ W/m$^2 \cdot$K | Thin insulation layer $3 < U < 1.8$ W/m$^2 \cdot$K | Medium insulation layer, $1.8 < U < 0.5$ W/m$^2 \cdot$K | Thick insulation layer, or $U < 0.5$ W/m$^2 \cdot$K |
| **Window (frame and glazing)** | Sold simple glazing frame $U_w > 6$ W/m$^2 \cdot$K | Old traditional single or double-glazing frame $6 < U_w < 3$ W/m$^2 \cdot$K | Newer traditional double-glazing frame $3 < U_w < 1.5$ W/m$^2 \cdot$K | Energy efficient double glazing fram $U_w < 1.5$ W/m$^2 \cdot$K |
| **Slab** | No insulation layer, or $U > 3$ W/m$^2 \cdot$K | Thin insulation layer $3 < U < 1$ W/m$^2 \cdot$K | Medium insulation layer, $1 < U < 0.7$ W/m$^2 \cdot$K | Thick insulation layer $U < 0.7$ W/m$^2 \cdot$K |

To create a pragmatic plan for potential renovation efforts, each energy improvement measure, whether related to the building envelope or systems, underwent a brief analysis to assess its potential interactions with other improvement measures in terms of comfort, building quality, conservation, energy performance, and environmental performance. The outcomes of these analyses were translated into conceptual diagrams, detailed in Section 3, "Results and Discussion". These diagrams illustrate the potential links or interactions with other building improvement measures for each energy improvement solution.

### 2.3.6. Listing of Potential Appropriate Renovation Measures

The central focus of the P-Renewal project is to provide adapted renovation solutions to improve the thermal comfort and energy performance of traditional residential buildings in Wallonia while preserving the heritage value and the integrity of buildings and integrating these solutions into a cohesive renovation strategy [66]. Several comprehensive studies have been conducted to achieve this goal.

Initially, a sizable inventory of technical improvement solutions, both for the building envelope and technical systems, was meticulously compiled. This inventory was developed based on the inputs from the literature [89–93], the analysis of exemplary case studies [15,16,41,58], and the collaboration of Walloon energy renovation experts and practitioners. For the energy improvement of the building envelope, the inventory of solutions was organized systematically and categorized wall by wall. Each wall was accompanied by a list of various insulation techniques, including options such as exterior and interior methods. Each insulation technique was examined with a focus on its advantages and disadvantages, feasibility, and consideration for the historic aspects of the building. For every insulation technique selected, the different possibilities in terms of construction techniques and building materials, as well as their procedures of implementation, were described in detail. The same methodology was applied to compile the inventory of solutions to improve or optimize the technical systems.

Then, for each study case, a more refined selection of energy improvement solutions was undertaken considering the architectural, spatial, constructive, and heritage-specific characteristics of each building, as well as their strengths and weaknesses in terms of energy and comfort.

Finally, decision trees were established for each envelope's wall and technical system to select an appropriate energy improvement solution according to the state of conservation of the wall, the system studied in its construction, or its technical features and heritage specificities. The trees were structured based on a series of closed questions requiring a "yes" or "no" answer. Depending on the answers, a path was generated to reach a specific energy improvement solution adapted to the studied building. Decision trees are presented in Section 3, "Results and Discussion".

### 2.3.7. Selection of Energy Renovation Strategies

Like Buda et al. [66], the P-Renewal project has defined "renovation strategies" as a combination of energy improvement solutions both on the building's envelope and on the technical systems. Four retrofitting strategies were proposed for each study case. Priority was assigned to energy interventions with a substantial impact on indoor comfort and energy performance. However, this priority was nuanced by other considerations, including preserving heritage value, cost-effectiveness, occupancy impact, and integrating alternative measures such as renewable energy production or building densification. The modulation of these four strategies across various indicators is detailed in Table 7. Strategy 1 can potentially be applied to many buildings as it mainly aims to insulate the roof and improve technical systems. It has no or very little influence on the heritage value and occupancy, and the financial investment is quite limited. Strategy 1′ is the same as Strategy 1, but it offers an alternative to "conventional" improvement measures, with either a contribution of renewable energy or a new spatial organization allowing a densification of the building. Strategy 2 improves the energy performance and comfort of the building by strengthening the insulation measures (facades and windows frame) and the efficiency of the systems while limiting the impacts on the heritage value as far as possible. The financial investment is higher while remaining moderate. However, the use

of the building is not always preserved during the work. Strategy 3 can potentially achieve the EPBlevel A, which the Walloon EPB regulation requires. The insulation measures are applied to all the walls of the envelope, the technical systems are all replaced, and a double-flow ventilation system is installed. However, it significantly influences heritage value, financial investment, and occupancy.

**Table 7.** Definition of the four strategies according to five indicators.

| | Criteria for Renovation Strategy's Evaluation | | | | |
|---|---|---|---|---|---|
| | **Energy Performance and Indoor Comfort** | **Heritage Value** | **Financial Investment** | **Building's Occupancy and Use** | **Alternatives Measures** |
| **Strategy 1** | Moderately improved | No impact | Limited | No impact | No |
| **Strategy 1′** | Moderately improved | No impact | Limited | No impact | Renewable energy production and/or building densification |
| **Strategy 2** | Improved | Low or moderate impact | Moderate | Low or moderate possible | Renewable energy production |
| **Strategy 3** | Significantly improved, EPB level A achieved | Significant impact | Significant | Significant impact | Renewable energy production |

The number of strategies proposed for each study case was deliberately limited to four to be able to assess in-depth both the overall energy improvement and the environmental and financial cost of the renovation measures as well as their technical reproducibility on the whole of the pre-war Walloon residential building.

### 2.3.8. Evaluation of Energy Renovation Strategies

The evaluation of renovation strategies has combined qualitative and quantitative assessments.

Qualitative assessment, conducted through a graphic scale, has examined three key aspects (Figure 5): the enhancement of indoor comfort, the potential risk of heritage value depreciation, and the impact of retrofitting on housing occupancy. While these criteria are crucial for evaluating the effectiveness of the proposed renovation strategies, they cannot be easily quantified by owners or professionals. These graphical scales facilitate a straightforward comparison between the current state of the building and the anticipated outcome with the proposed improvement solutions.

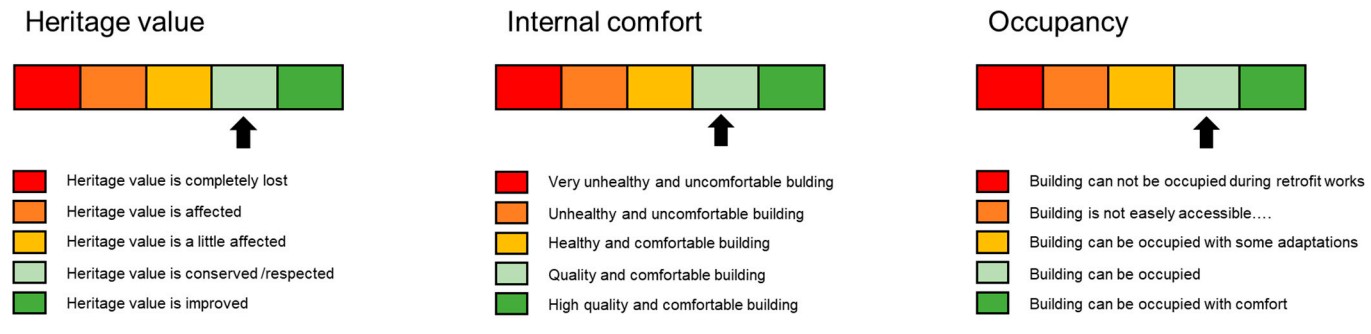

**Figure 5.** Qualitative criteria used for the evaluation of renovation strategies [72].

Quantitative assessment has concentrated on four key indicators: the U-value of each renovated wall or element, the percentage of energy savings achieved, the evaluation of the environmental impact of materials used for energy improvement, and the economic cost of each renovation strategy.

The U-values of the renovated walls were calculated via the EPB software v10 developed by the Walloon Region, according to the selected improvement solution and the insulation materials used (based on their conductivity and thickness).

The energy savings of each renovation strategy have been assessed with the dynamic energy models developed with Design Builder software, knowing that each energy improvement solution has a specific impact on global energy savings. A baseline scenario for the heating system settings is required to test the improvement solutions: the heating demand varies according to the season, the hours of occupancy of the building (different on weekdays and weekends), and the rooms. Energy improvement solutions can then be compared to the baseline scenario. An energy gain (in kwh per year) is then calculated for each solution and each case study. Several different alternatives were also proposed for the choice of insulation. Finally, strategies combining several improvement solutions were proposed and compared in terms of final energy savings.

The environmental impact of each energy improvement solution has been assessed via the TOTEM software (version of March 2020) [94]. The compositions of each existing wall and energy improvement solution were introduced in the software. Among the 17 environmental indicators provided using TOTEM, the assessment study focused on (1) the embodied energy within the life cycle of new building materials or the depletion of abiotic resources—fossil fuels, measured in $MJ/m^2$ of wall; and (2) greenhouse gases emitted/generated throughout the life cycle of new building materials or global warming potential, measured in kg $CO_2$ eq./$m^2$ of wall.

The economic cost of one square meter of each energy improvement measure was calculated based on several documents from the CO-ZEB studies [17]. The objective of these studies was to identify a cost-effective energy renovation strategy and the optimum cost for energy retrofitting of the envelope adapted to the Walloon dwelling stock. As an example, the results achieved by the four renovation strategies are presented for study cases 4 and 5 in Table 8. It is important to mention that only envelope insulation measures have been included in these strategies, considering various choices of insulating materials.

**Table 8.** Results of the four renovation strategies in terms of annual energy savings, environmental impact, and investment costs.

| | Annual Energy Saving for Heating. kWh | Environmental Impact of Insulation Measures | | Investment Costs for Renovation Works Euros (€) |
|---|---|---|---|---|
| | | Grey Energy Demand kWh | Global Warming Potential kg $CO_2$ equ. | |
| **Study case 04,** with a net annual heating demand of 19,195 kWh for a floor area of 137 $m^2$ | | | | |
| Strategy 01 | Between 2304 and 2363 **(13%)** | Between 35,603 and 42,167 | Between 8521 and 9583 | Around EUR 12,000 |
| Strategy 02 | Between 2304 and 2363 **(13%)** | Between 35,603 and 42,167 | Between 8521 and 9583 | Around EUR 12,000 |
| Strategy 03 | Between 5010 et 5103 **(27%)** | Between 63,144 and 67,970 | Between 15,505 and 18,577 | Around EUR 47,000 |
| Strategy 04 | Between 5761 and 5872 **(31%)** | Between 62,599 and 76,928 | Between 15,867 and 18,435 | Around EUR 49,000 |

**Table 8.** *Cont.*

| | Annual Energy Saving for Heating. kWh | Environmental Impact of Insulation Measures | | Investment Costs for Renovation Works Euros (€) |
| | | Grey Energy Demand kWh | Global Warming Potential kg CO$_2$ equ. | |
|---|---|---|---|---|
| Study case 05, with a net annual heating demand of 46 160 kWh for a floor area of 215 m$^2$ | | | | |
| Strategy 01 | Between 2376 and 2693 **(5%)** | Between 70,398 and 80,405 | Between 14,362 and 16,686 | Around EUR 42,000 |
| Strategy 02 | Between 3598 and 3962 **(8%)** | Between 62,451 and 71,677 | Between 15,303 and 18,393 | Around EUR 46,000 |
| Strategy 03 | Between 16,339 and 16,771 **(36%)** | Between 107,818 and 171,472 | Between 39,453 and 46,610 | Around EUR 76,000 |
| Strategy 04 | Between 17,114 and 17,546 **(38%)** | Between 111,608 and 175,858 | Between 40,703 and 47,976 | Around EUR 78,000 |

The qualitative and quantitative assessments have demonstrated that the four energy renovation strategies significantly enhance the comfort and energy efficiency of the studied buildings. Importantly, three of these strategies successfully maintain the buildings' heritage characteristics and limit environmental and cost impacts. Notably, improvements in thermal comfort have been achieved by effectively mitigating the radiative "cold wall" effect, even though some energy improvement solutions do not consistently meet the maximum U-value required by the Walloon EPB regulation. However, while aligning with regulatory requirements, one of the four strategies involves a compromise on heritage value and incurs a relatively high environmental impact due to the introduction of new materials. Moreover, the implementation of this strategy carries substantial financial costs. Those aspects reduce its feasibility for widespread replication on a large scale compared to the first three strategies.

## 3. Results and Discussion

The methodological steps of the P-Renewal project have led to a more structured and detailed reflexive renovation planning process, enabling building owners and/or renovation professionals to identify the reference building type, document and characterize their building's specificities, and appropriately select energy improvement solutions for their integration into a renovation strategy. Furthermore, various outcomes have emerged from the studies and analyses conducted during the project, including substantial data and technical information. Crucially, the project has yielded diverse decision-making tools that support the reflective process, as suggested by Leijonhufvud, Broström, Buda, et al. [69,70].

Hence, while the P-Renewal project aligns with the objectives of the standard EN16883 and addresses the same target audience, the technical data, information, and tools generated by the project should be viewed primarily as significant support for owners and practitioners in their reflective renovation processes. Additionally, these outcomes can be considered robust scientific contributions, fostering the enhancement of the usability of the standard and the evolution of energy performance standards for historic buildings.

### 3.1. Reflexive Renovation Planning Process Based on Real Study Cases

As shown in Figure 6, the reflexive planning process of P-Renewal aligns with the objectives and planning process outlined in the standard EN 16883. While both processes were conceived through a multi-criteria approach that incorporates energy improvement and heritage value, a notable scientific contribution of the P-Renewal project lies in its

utilization of real study cases and consideration of the requirements of renovation professionals. This bottom-up and integrated approach has served to substantiate and validate the development of a reflexive process. Moreover, it has contributed to facilitating access to collected data and integrating technical information on various building types and renovation solutions. The discussions with the User Group played a crucial role in considering the constraints and requirements of the field. They also fostered the development of a shared understanding among all stakeholders, from researchers to professionals. This collective understanding clarified what is necessary, feasible, and ideal during the renovation process. Importantly, it led to the proposal of retrofitting measures and strategies that were widely accepted by all involved parties.

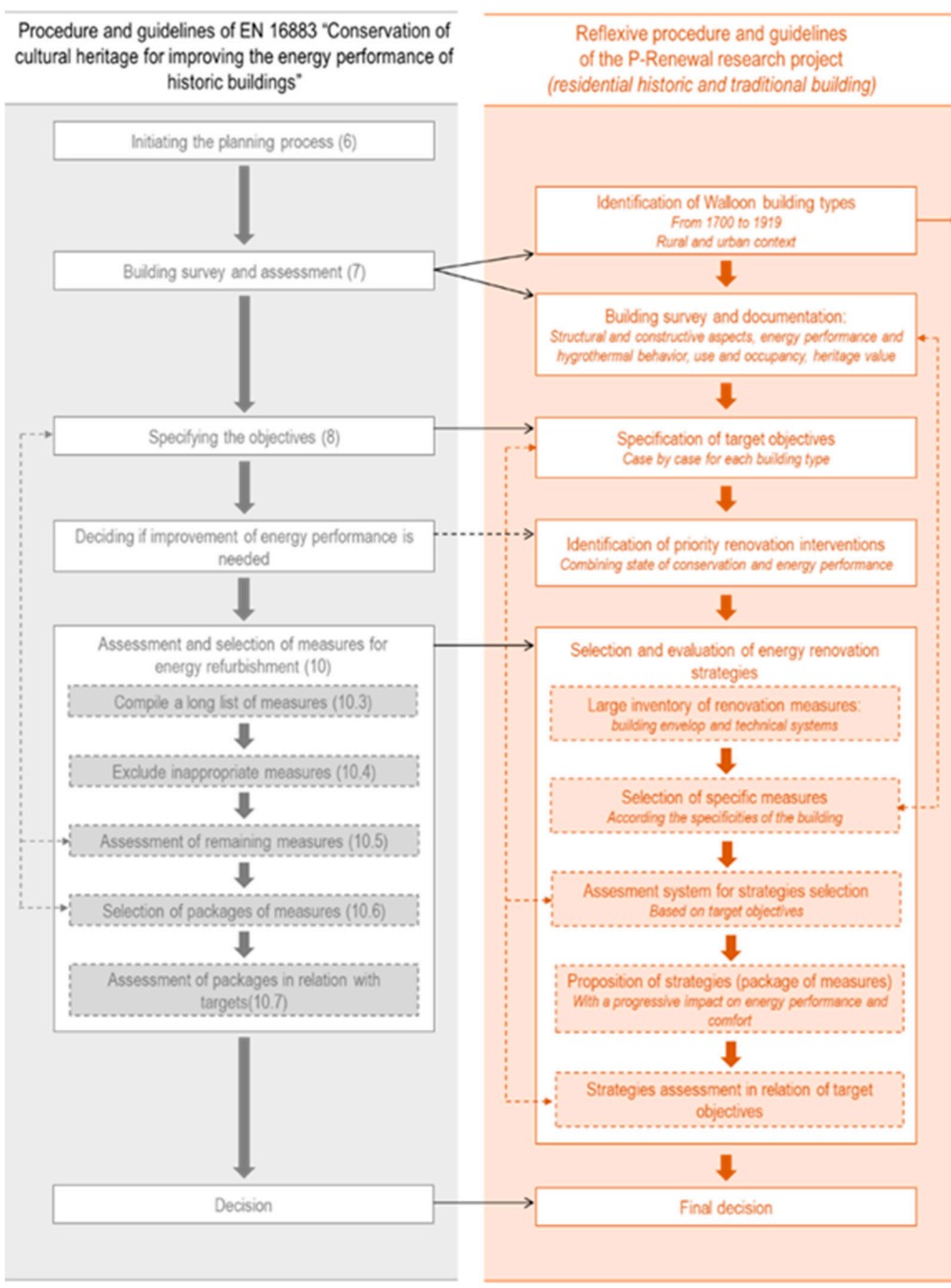

**Figure 6.** Comparison between the two flow charts: standard (in grey) and P-Renewal project (in orange).

The P-Renewal project has also proposed several new methodological steps that provide more details and strengthen the planning process. This is particularly the case for step 7 of the standard process «Building survey and assessment» for which the P-Renewal project provides a step of identification of the built type before the step of survey and documentation of the studied building. This is also the case for step 10 of the standard process "Assessment and selection of measures for energy renovation", for which the P-Renewal project provides a methodological approach to select renovation solutions in adequacy with the specificities of the studied building as well as a set of evaluation criteria for selecting renovation strategies. These new steps into the reflexive process are supported by decision support tools described in Section 3.2.

### 3.2. Decision-Making Tools and Technical Data Supporting the Renovation Planning Process

Unlike the standard EN 16883's procedure and guidelines, the reflexive planning process proposed by the P-Renewal project is supported both by data collected on five study cases representative of the Walloon built stock and technical information compiled from the literature and discussions with User Group experts. The process is further reinforced by a range of tools developed within the research project, facilitating the documentation and characterization of buildings, selecting and evaluating suitable renovation solutions, and integrating them into a comprehensive renovation strategy.

As depicted in Figure 7, the scientific interest in the P-Renewal project extends beyond the reflexive process, primarily residing in the decision-making tools offered to owners and practitioners to enhance their renovation planning. These contributions serve the dual purpose of guiding the execution of each step, facilitating the collection of requisite data for progression to subsequent stages, and providing the necessary technical information for the selection of appropriate renovation interventions and measures. The different decision-making tools and other outcomes are presented briefly below, following the steps of the reflexive process.

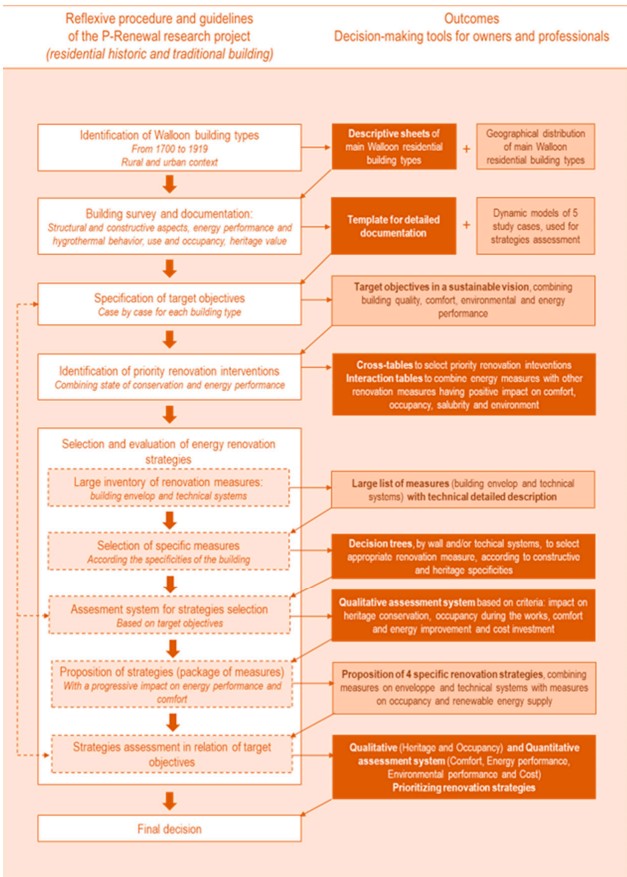

**Figure 7.** Reflexive planning process and outcomes of the research project of P-Renewal.

### 3.2.1. Typological Analysis and Identification of Pre-War Walloon Building Types

The typological analysis serves to address, albeit partially, the existing gap in data and knowledge concerning the material and construction aspects of the traditional residential building stock in Wallonia. This is achieved through the characterization of each type, encompassing its architectural, structural, spatial, and heritage-specific features. Furthermore, estimating the representativeness of the main building types has enabled the identification of priority structures (in terms of quantity) where energy renovation will exert a substantial influence on the region's energy balance.

The analysis has yielded comprehensive and visually supported descriptive sheets for the seven main pre-war Walloon residential building types. These sheets provide a detailed presentation of each building type, encompassing its historical evolution, dimensions, forms and volumes, spatial organization, relationship to public space, and constructive systems and building materials. Additionally, the descriptive sheets outline the most encountered heritage specificities and reference-built examples.

In contrast to the procedure and guidelines of standard EN 16883, these descriptive sheets offer a streamlined and readily accessible source of data and information on pre-war residential building types. This facilitates a quick and straightforward identification of the specific type to which the studied building belongs. Moreover, these sheets enable users to gain awareness of the potentially constructive and heritage specificities of their building, factors that could significantly impact the selection of energy improvement solutions. A first enhancement to the standard could involve incorporating this type of descriptive sheet, considering that such sheets can be readily generated at both the European and regional levels. This could be achieved by drawing upon various studies conducted on historical building types, such as those undertaken in projects like [13,55].

### 3.2.2. Selection and Documentation of Study Cases

This phase has yielded a significant volume of data for each study case, enabling the identification of all the constraints that need to be addressed and the opportunities that can be leveraged within the renovation process. These data have been compiled and documented in a report that outlines the identity of each study case and reveals their strengths and weaknesses in terms of energy efficiency, comfort, and state of conservation.

The data gathered during the building's survey have facilitated the creation and organization of straightforward questionnaires and templates designed for building surveys and diagnostics. Tailored for a non-expert audience, these questionnaires enable the provision of a general description of the studied building by collecting data on each wall, encompassing material and construction aspects, state of conservation, and heritage features. While these data do not replace a comprehensive diagnostic and in-depth analysis of the building's energy performance, they present a holistic and realistic overview of the structure and its specific characteristics. The questionnaires were utilized by Buildwise to create the online tool Renocheck [95], enabling a quick and effective assessment of a building's state of conservation before undergoing renovation.

Moreover, the table summarizing the building's heritage value streamlines the identification of specific heritage elements that ideally should be preserved during renovation interventions. As emphasized by standard EN 19883, identifying and classifying these heritage elements must be considered a fundamental prerequisite for establishing target objectives and selecting the most suitable intervention strategies.

In contrast to the procedure and guidelines of standard EN 16883, these questionnaires offer a strong technical basis for documenting and analyzing an existing traditional or historical building before considering renovation measures. This kind of questionnaire could be appended to the standard as an illustrative example of the procedure to be followed for the documentation and surveying phase of a building.

Furthermore, the P-Renewal project has provided information on how pre-war historic residential buildings can be adapted for alternative functions and/or increase their housing density. This scientific contribution should be recognized as a valuable input for enhancing standard EN 16883. It is noteworthy that numerous European cities and rural areas are currently grappling with an increasing demand for housing, and the densification of buildings can play a role in the energy transition of cities [96,97].

### 3.2.3. Identification of Priority Renovation Interventions

This step has yielded two decision-making tools: cross-tables, which enable the identification of the priority level for renovation interventions for each wall of the envelope, and conceptual diagrams, which facilitate the combination of energy improvement measures with other renovation measures that positively impact comfort, occupancy, housing quality, and the environment. As an illustration, Table 9 presents the cross-table for pitched roofs. In this case, the roof's insulation is considered a priority due to the absence of insulation and under-roof (rainproof protection) and the heat losses given by the high air infiltration rate.

**Table 9.** Pitched roof—Cross-table for identification of renovation intervention's priority level.

| | | | State of Conservation | | | |
|---|---|---|---|---|---|---|
| | | | **1** | **2** | **3** | **4** |
| | | | **Priority** | **Necessary** | **Possible** | **Nonpriority** |
| Energy performance | 1 | Very energy intensive $U \geq 1.5$ W/m²K | Priority intervention | Priority intervention | Necessary intervention | ✕ |
| | 2 | Energy intensive $1.5 \geq U \geq 1$ W/m²K | Priority intervention | Priority intervention | Necessary intervention | ✕ |
| | 3 | Moderately energy intensive $1 \geq U \geq 0.4$ W/m²K | Priority intervention | Priority intervention | Possible intervention | Nonpriority intervention |
| | 4 | Energy efficient $U \leq 0.4$ W/m²K | ✕ | ✕ | Nonpriority intervention | Nonpriority intervention |

Conceptual diagrams, as illustrated in Figure 8, have been developed for various energy improvement solutions. The energy improvement solution, in this instance, the insulation of the roof, is regarded as the starting point of the energy renovation process, given its priority in addressing energy efficiency. The uncolored boxes in the diagrams depict elements (building and technical equipment) that directly interact with the roof insulation on technical, health, and/or economic levels. These diagrams can assist owners and/or practitioners in crafting a cohesive set of interconnected energy interventions for a specific wall/space or as part of a broader renovation plan. They contribute to enhancing the efficiency of the renovation planning process by minimizing time and cost, and they can also be viewed as valuable support in aligning with the roadmaps advocated by the European Union as part of long-term renovation strategies.

These two decision-making tools offer valuable scientific and technical input compared to the standard EN 16883. Additionally, they serve as practical support for the renovation planning process by enabling non-technical users to select and integrate energy improvement measures with other renovation measures. A potential next step in the evolution of the standard could involve incorporating these decision-making tools, with appropriate adjustments based on the energy performance requirements specific to each European country.

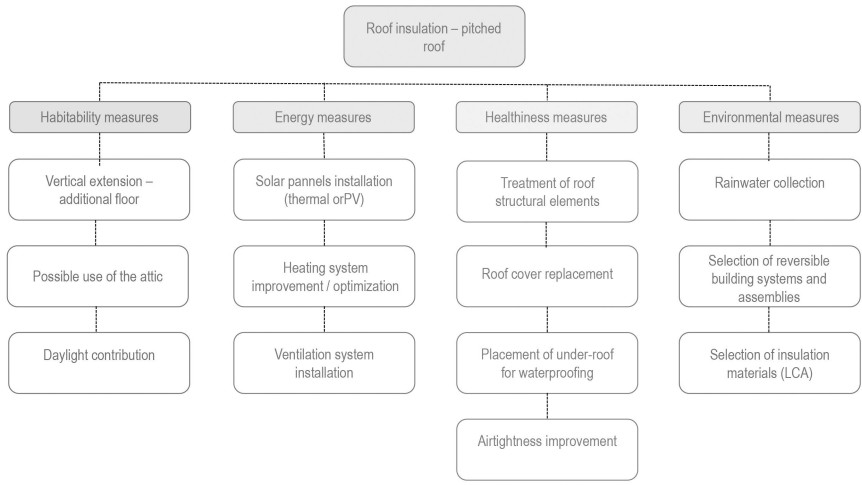

**Figure 8.** Potential interactions between an energy improvement solution on a pitched roof and other improvement measures.

### 3.2.4. Listing of Potential Appropriate Renovation Measures

The detailed and illustrated inventory developed within the P-Renewal project should be regarded as a significant contribution. It provides essential knowledge on various technical solutions to enhance the energy performance of walls or technical systems. This knowledge is crucial in motivating property owners to enhance the energy efficiency of their buildings, aligning more effectively with Europe's zero-carbon objectives. Considering its value, this inventory could undergo revision, improvement, and/or expansion through collaboration with a panel of international experts before being appended to the standard EN 16883.

Furthermore, decision trees should be recognized as a valuable technical contribution, assisting owners or professionals in selecting energy improvement measures tailored to the specificities of the studied building. A potential third evolution of the standard could involve incorporating these decision trees, with adjustments made according to the heritage and constructive specificities of different building types, drawing upon various studies conducted on historical structures [13,55]. As an example, the decision tree for enhancing the energy performance of a pitched roof is presented in Figure 9.

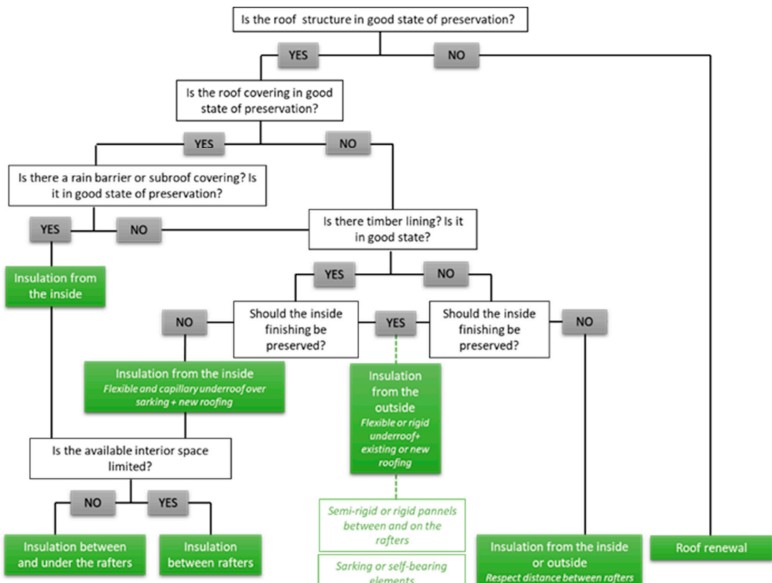

**Figure 9.** Decision tree for the energy improvement of the pitched roof.

3.2.5. Evaluation of Renovation Strategies

The P-Renewal project has introduced both qualitative and quantitative assessment systems for evaluating energy renovation strategies. While the quantitative assessment depends on the specific energy regulation of each European country, on diverse climates across the European territory, as well as on specific environmental assessment tools and financial systems for building materials, the qualitative assessment provides the capability to swiftly evaluate the advantages or drawbacks resulting from an energy improvement solution and/or a renovation strategy concerning thermal comfort, building occupancy, and heritage value. This qualitative evaluation system could also lay the foundation for developing a more in-depth system, offering a comprehensive and holistic evaluation of renovation operations on traditional and historic buildings.

It is crucial to acknowledge that there is currently no consensus on the assessment of heritage value [98] despite various methods proposed in different projects, including the EFFESUS project [55]. The absence of a consensus and a common framework poses a challenge and does not facilitate the widespread adoption of the standard by professionals.

*3.3. Web tool and Recommendations Sheets*

The entire reflexive planning process, along with technical data and tools, has been incorporated into a web tool to effectively address the needs of the Walloon renovation sector, particularly owners and practitioners. The web tool is currently in development and is scheduled to be made public in the coming months. Additionally, recommendation sheets have been drafted for owners of the main seven pre-war residential building types. These sheets guide owners through the step-by-step reflexive process, utilizing the developed tools to provide justification or facilitate discussions on the choice of energy improvement solutions.

**4. Conclusions and Perspectives**

Embracing a bottom-up, inclusive, and multi-criteria approach, the P-Renewal project was designed to assist in the energy renovation of historic Walloon residential buildings with heritage value. It encourages various stakeholders in the renovation process to foster a holistic and comprehensive perspective on the numerous challenges involved. In contrast to the standard, which renovation professionals perceive as a set of obligatory rules, the P-Renewal project and its outcomes are regarded as practical tools and support to ensure compliance with legal energy requirements and more sustainable renovation objectives.

The reflexive process developed by the project can serve as an exemplary model and complement existing local and European regulations. It offers valuable assistance to building owners and practitioners in formulating appropriate renovation strategies. Moreover, the additional steps introduced by the P-Renewal project could be incorporated into the future evolution of the standard EN 16883 to enhance the renovation planning procedure.

The input of data and technical information, as well as decision-making tools provided by the project, act as practical support for renovation professionals and practitioners, facilitating the seamless integration of the renovation planning procedure outlined in this standard. These outcomes can be easily adapted to various contexts and building types, drawing upon the wealth of studies conducted on European built types and energy improvement measures.

The four renovation strategies presented in the project underscore the challenges associated with meeting the thermal transmission coefficient "U" requirements for each wall of the envelope. This task involves balancing the need for energy improvements with the preservation of heritage characteristics and minimizing the overall environmental impact. While some walls can easily accommodate energy enhancements and the required thickness of thermal insulation, others present greater complexities and difficulties in terms of improvement. It is evident that a shift in energy regulations is imperative. Rather than assessing each wall's performance, a historic building's energy performance should be evaluated within a defined volume bounded by a set of walls, which may or may not meet or even exceed the specified requirements. This concept can be extended to technical

systems, where, in the case of historic buildings, their integration may be complex, even though they can have a considerable beneficial influence—ventilation systems being one example. Moreover, the densification of building occupancy and the incorporation of diverse functions within the same building should also be considered. Both aspects play a significant role in the overall energy performance of a building.

For traditional and historic buildings, particularly those not listed or on an inventory list, it is both urgent and essential to collaboratively advance at the European level in the assessment of heritage value and the evaluation of the impact of renovation measures on heritage specificities and elements. Without such progress, there is a significant risk that our landscapes and cities may lose a substantial portion of their historical, landscape, and cultural quality. This quality must be recognized as a tangible heritage to be passed down to future generations.

Finally, European energy regulations must progress towards a more comprehensive and sustainable vision that considers both the grey energy and carbon footprint associated with the life cycle of materials [99,100] utilized to enhance building performance. Additionally, attention should be given to the waste generated by increasing renovation operations [101,102]. Combining energy performance and environmental assessment, as demonstrated by the French government in 2022 [103], could incentivize the renovation sector and professionals to adopt more reversible techniques (circular design) and increase the use of bio-based materials for insulation.

In conclusion, the P-Renewal project has demonstrated that heritage value and energy performance should not be perceived as antagonistic objectives; rather, they should be considered as complementary goals. Simultaneously addressing spatial use, internal comfort, environmental performance, and circular economy issues is essential. Indeed, it is possible to improve energy performance and occupants' comfort while preserving heritage values, although this requires a proper consideration of requirements and regulations, for example, in terms of the thermal transmittance of individual building components. Historic buildings possess inherent assets and specificities that, once acknowledged, should guide the priorities of each renovation project. To achieve this goal, a detailed analysis of existing materials and components should be undertaken to better characterize the thermal and hygrometric behaviors of walls and validate proposed improvement solutions. However, the generalization and transfer of solutions demand substantial data from available case studies, emphasizing the need for extensive research in the future.

**Author Contributions:** Writing—original draft preparation, S.T.; writing—review and editing, S.T. and D.S.; visualization, S.T.; supervision, S.T. and D.S.; project administration, S.T. and D.S.; funding acquisition, S.T. All authors have read and agreed to the published version of the manuscript.

**Funding:** The P-Renewal research project was funded by the SPW-DG4 (Direction générale opérationnelle de l'aménagement du territoire, du logement, du patrimoine et de l'énergie du Service public de Wallonie)—Grants n°1650366 (UCLouvain) and n°1650367 (Buildwise).

**Data Availability Statement:** The authors point out that all data, reports and results of the P-Renewal project were written in French and not translated into English. The web tool, also in French, is being finalized. It will include all the results and decision-making tools presented in the manuscript. All data presented in this manuscript will be available on request from the two auteurs, and with the permission of the research partner Buildwise.

**Acknowledgments:** The authors acknowledge the researchers of the Buildwise Unit Building Performances & Renovation, especially Michael de Bouw, Samuel Dubois, Julie Desarnaud, and Yves Vanhellemont, for their participation and contribution to the P-Renewal project and especially Samuel Dubois for its pictures and paper's reviewing. Furthermore, not being English native speakers, the authors used IA Technologies (Chat GPT) to improve the English linguistics of some parts of the manuscript (materials and methods, results, and discussions). A first translation was made by the authors, and then some passages of the manuscript were improved using Chat GPT.

**Conflicts of Interest:** The authors declare no conflicts of interest.

## Notes

1. Long-term renovation strategies—https://energy.ec.europa.eu/topics/energy-efficiency/energy-efficient-buildings/long-term-renovation-strategies_en (accessed on 5 March 2024)

2. The percentages highlighted in the text come from several sources which relied on data from Eurostat and/or surveys carried out by the BPIE (for Europe) and from Belgian statistic database Statbel (for Wallonia).

3. The definition proposed by Webb corresponds to the pre-war Walloon building types studied by the authors. These buildings were built before 1919, they are built with traditional construction techniques and permeable materials (traditional). They are more than 50 years old, retain the integrity of the physical features that existed during the historical period of the property, and have significance in terms of probative, historical, landscape or community value (historic).

4. Troi used statistical data from the "Bulletin of Housing Statistics for Europe and North America 2004". These bulletins are regularly published by the United Nations Economic Commission for Europe (UNECE).

5. Listed properties are selected from the public domain on the basis of their local heritage value. This selection is based on various criteria and interests, used alone or in combination, which serve as guidelines for listing, and guarantee the objectivity of the choices made. The criteria used are (84. SPW-DGO4. *Inventaire du patrimoine culturel immobilier*. Available online: http://spw.wallonie.be/dgo4/site_ipic/ (accessed on 5 March 2024)):

- Authenticity: the property's function and use, form and materials, and surroundings correspond to its original state.
- Integrity: the property is homogeneous and coherent. The original functions are still clearly identifiable, despite the change of use.
- Rarity: the property is locally unique, rare or exceptional, even if fragmentary, in terms of its typology, style, dating or social or historical interest.
- Typology: the property possesses architectural characteristics linked to a specific function.

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
