# Peer review of "P-Renewal Project: A Reflexive Contribution to the Evolution of Energy Performance Standards for the Renovation of Historic Buildings"

_heritage, doi:10.3390/heritage7030074_

Round 1
Reviewer 1 Report
Comments and Suggestions for Authors
Although the objectives of the research are of interest and the research methodology is well explained, there is no supporting data from the case studies on which it is declared that the study was carried out. Lacking case study data, the study remains at a theoretical level. The validity of the methodology must be demonstrated by application to case studies.
The cited inventory developed within the P-Renewal project should be reported as to better explain the methodology.
In the introduction there are various repetitions of the purpose of the standard and of the study, to be removed.
In table 3, specify how the ranges of thermal transmittance values of the various elements were established.
Explain why the second strategy in table 4 is indicated as “strategy 1'” and not as “Strategy 2”.
Report in figure 5 the indication of reference [70] where the same graphs have already been published
Author Response
Thank you very much for taking the time to review this manuscript. Please find the detailed responses below and the corresponding revisions/corrections highlighted/in track changes in the re-submitted files. Concerning your general evaluation, we clarified comprehensively the presentation of the questions posed by the research, the methodological approach, and the results and we improved the presentation of the results by providing data.
You'll find below our responses to your comments, point by point
Comment 1: Although the objectives of the research are of interest and the research methodology is well explained, there is no supporting data from the case studies on which it is declared that the study was carried out. Lacking case study data, the study remains at a theoretical level. The validity of the methodology must be demonstrated by application to case studies.
RESPONSE
Thank you for pointing this out. We agree with this comment. Therefore, three tables have been incorporated into section 2 “Methodology”, point 2.3.3 (lines 444, 461 and 500) to comprehensively describe the analyses conducted on the study cases, along with pertinent data on the existing energy performance. In point 2.3.8 (line 684), we also have included a table with data on energy performance achieved in each strategy. We believe that these different elements allow to better understand the validity of the global method and the reflexive process.
Comment 2: The cited inventory developed within the P-Renewal project should be reported as to better explain the methodology.
RESPONSE
We agree with this comment. We have appended a brief passage to section 2 “Methodology”, point 2.3.6 (page 16, line 584) to provide further explanation on the methodology used for creating the inventories.
Comment 3: In the introduction there are various repetitions of the purpose of the standard and of the study, to be removed.
RESPONSE:
Agree. We have made some overall improvements to the wording in the section 1 “Introduction” to minimize the repetition of the terms “P-Renewal project” and “EN standard”. We have added some sentences in the second paragraph (page 1, line 43) to better explain the contributions of the P-Renewal project compared to the standard. We also have included a sentence in point 1.2 (page 6, line 263), to explicitly articulate the objectives of the P-Renewal project. With this addition, we believe that any redundancy has been eliminated.
Comment 4: In table 3, specify how the ranges of thermal transmittance values of the various elements were established.
RESPONSE:
Thank you for pointing this out. A short explanation of how the ranking was established has been added in point 2.3.5 (page 15, line 541) before the tables.
Comment 5: Explain why the second strategy in table 4 is indicated as “strategy 1'” and not as “Strategy 2”.
RESPONSE
Thank you for pointing this out. A short explanation has been added in point 2.3.7 (page 17, line 614) above the table. Each strategy is now briefly described.
Comment 6: Report in figure 5 the indication of reference [70] where the same graphs have already been published.
RESPOND
Yes, you’re right. We forgot to mention it during the first submission (as well as the figure 4). This will not be mentioned when submitting the revised manuscript. Those two figures were used in a paper written in French in 2020 and in a conference paper in 2019 by the same authors. Those two papers has as objective to describe briefly the main methodological steps of the research project P-Renewal without referring or comparing with the reflexive process proposed by the standard EN 16883. Those papers did not present or discuss the research results, or the decision-making tools developed within the project. To be transparent, we have referenced these papers in the caption of each illustration.

Reviewer 2 Report
Comments and Suggestions for Authors
The paper is well-written and interesting, worthy of publication.
There are a few areas where authors might consider making improvements for clarity and precision.
1. Specify some key differences or specific points where the P-Renewal approach deviates from the guidelines outlined in EN 16883. This will provide a more detailed understanding of the unique features of the P-Renewal reflexive process.
2. Mention the specific locations or names of the case studies in Wallonia. This can add concreteness to your description and help readers connect better with the examples.
3. Provide a summary or examples of the "advanced scientific and technical information and decision-making tools" mentioned in the text.
3. Elaborate on the role of the construction sector in promoting a more sustainable and carbon-neutral economic system. Consider providing examples or statistics to support this claim.
4. Clarify the term "long-term renovation strategies." Explain what these strategies entail and how they contribute to making existing building stocks highly energy-efficient by 2050.
5. Provide the source of the statistics mentioned, especially when referencing percentages of building stocks in Wallonia and Europe.
6. Specify the types of energy performance improvements that are needed for the existing buildings, especially for those built before 1960.
7. Define what is meant by "historic and traditional buildings" and provide examples.
8. When mentioning various research projects, provide a brief description of each project's focus or objectives.
9. When presenting quantitative data, such as the potential annual CO2 savings, consider providing the source of this information to enhance credibility. Additionally, ensure that the units (e.g., Mt of CO2) are explained for clarity.
10. Clarify the term "buffer spaces" in the context of thermal and hygrothermal behavior.
11. Elaborate on the challenges or considerations associated with generalizing renovation solutions from one case to another, especially regarding the preservation, structural composition, climatic conditions, and occupant behavior.
12. Ensure consistency in the use of terms such as "renovation" and "refurbishment." Use these terms consistently throughout the text to avoid confusion.
13. In the section discussing the survey and diagnosis stage, the authors mention focusing on the "four dimensions of the project." It would be helpful to state what these dimensions are, providing clarity explicitly.
14. While it's mentioned that the User Group includes architects, renovation companies, energy efficiency experts, and heritage refurbishment experts, authors might elaborate on how their input influenced decision-making. How did their expertise contribute to the project?
15. In the Multidisciplinary Approach section, authors mention criteria such as the potential occupancy of the building, comfort improvement, conservation of heritage value, financial cost, and overall environmental impact. Consider briefly specifying how these criteria were weighted or prioritized in decision-making.
16. When discussing the analysis of heritage significance, consider providing a brief explanation of the four criteria (authenticity, integrity, rarity, and representativeness) and how they were applied.
17. Improve references with the following: I.J., Gil Crespo et al., “Climatic analysis methodology of vernacular architecture”, in Vernacular Architecture: Towards a Sustainable Future, pp. 327–32, Taylor & Francis Group, London, 2015; F. Ribera, P. Cucco, I.J.G. Crespo, Energy efficiency features in italian and spanish traditional dwellings, SUSTAINABLE MEDITERRANEAN CONSTRUCTION, 12/2020, pp. 178-183; J. Fernandes et al., The influence of the Mediterranean climate on vernacular architecture: a comparative analysis between the vernacular responsive architecture of southern Portugal and north of Egypt, in Proceedings of the World Sustainable buildings SB14, Barcelona, Spain, 28-30 October 2014; P. Cucco, Dalla Conservazione Integrata di Amsterdam (1975) all’Integrated Approach to Cultural Heritage (2020). Nuove prospettive nello scenario di cambiamenti globali, EDA. ESEMPI DI ARCHITETTURA, 2/2020, pp. 1-10.
Author Response
Thank you very much for taking the time to review this manuscript. Please find the detailed responses below and the corresponding revisions/corrections highlighted/in track changes in the re-submitted files. We tried to clarify comprehensively the presentation of the questions posed by the research, the methodological approach, and the results. You’ll find below, our responses to your comments, point by point.
Comment 1: Specify some key differences or specific points where the P-Renewal approach deviates from the guidelines outlined in EN 16883. This will provide a more detailed understanding of the unique features of the P-Renewal reflexive process.
RESPONSE:
Thank you for pointing this out. We added some explanations on this point, in the section 1 “Introduction” (page 1, line 43 and page 6, line 263). Specific points of P-Renewal project are presented and discussed in section 3 “Results and discussion”.
Comment 2: Mention the specific locations or names of the case studies in Wallonia. This can add concreteness to your description and help readers connect better with the examples.
RESPONSE
Unfortunately, according to the EU General Data Protection Regulation (GDPR), we are not authorized to mention the name of buildings owners and the address of the building. Only the city or village where the building is located can be mentioned. However, we add some information below the figure 3, in point 2.3.2 (page 10, line 423).
Comment 3: Provide a summary or examples of the "advanced scientific and technical information and decision-making tools" mentioned in the text.
RESPONSE
Agree. We added some examples in the section 1 “Introduction” (second paragraph, page 1, line 43), and we have detailed these different inputs in the section 3 “Results and discussion”, point 3.2.
Comment 4: Elaborate on the role of the construction sector in promoting a more sustainable and carbon-neutral economic system. Consider providing examples or statistics to support this claim.
RESPONSE
We added a sentence with statistics in the section 1 “Introduction” (third paragraph, page 2, line 59) to explain the role of the construction sector in promoting a more sustainable and carbon-neutral renovation. References have also been expanded, with quantified data.
Comment 5: Clarify the term "long-term renovation strategies." Explain what these strategies entail and how they contribute to making existing building stocks highly energy-efficient by 2050.
RESPONSE
We think that the term “long-term renovation strategies” is well known in the energy renovation sector (both scientifically and professionally). This is a European obligation. That is why we added a footnote (page 2, line 69) with a reference to the European Commission website. If you think more information is needed, we could add the text below:
“Those national and or regional strategies must be integrated in energy and climate plan (NECPS) of each EU country member. They must include an overview of (1) the national building stock policies and actions to stimulate a cost-effective deep renovation, of (2) worst performing buildings, split-incentive dilemmas, market failures… and of (3) national initiatives to promote smart technologies and skills in the construction and energy efficiency sectors. They also must propose a roadmap with measures and measurable progress indicators and indicative milestones, and an estimation of the expected energy savings and the contribution to the Union's energy efficiency target”.
Comment 6: Provide the source of the statistics mentioned, especially when referencing percentages of building stocks in Wallonia and Europe.
RESPONSE
When percentages and statistics are mentioned, we have referenced the scientific reports or publications that provide these percentages/statistics. These reports and/or publications used specific statistical databases and/or surveys that were referenced in footnote. See section 1 “Introduction” (page 2, line 74 and page 3, line 92).
Comment 7: Specify the types of energy performance improvements that are needed for the existing buildings, especially for those built before 1960.
RESPONSE
Agree. We added some explanation about energy improvement measures that are needed on existing buildings built before 1960, in section “Introduction” (page 2, last paragraph, line 82). Significant energy savings are already possible with the implementation of a few energy improvement measures, but to achieve zero energy level, it is necessary to it is necessary to work both on the insulation and airtightness of the envelope, the integration of a ventilation system and the replacement of certain technical elements or equipment influencing the energy demand. Some examples of improvement measures have been added along with the expected gain, as referenced in the text.
Comment 8: Define what is meant by "historic and traditional buildings" and provide examples.
RESPONSE
Thank you for pointing this out. The definition proposed by Webb (reference mentioned in the manuscript) corresponds to the pre-war Walloon building types studied by the authors. These buildings were built before 1919, they are built with traditional construction techniques and permeable materials (traditional). They are more than 50 years old, retain the integrity of the physical features that existed during the historical period of the property, and have significance in terms of probative, historical, landscape or community value (historic). This explanation has been added as a footnote in the section 1 “Introduction” (page 3, line 89) so as not to burden the text.
Comment 9: When mentioning various research projects, provide a brief description of each project’s focus or objectives.
RESPONSE
While we understand the value of the request, we think providing a short description of each research project mentioned in the background may greatly lengthen this section, even though it is not the heart of the article. Each research project is well referenced in the section “References” with the URL of the website (if there is one). But, if you think it’s needed, we could add an appendix with this information.
Comment 10: When presenting quantitative data, such as the potential annual CO2 savings, consider providing the source of this information to enhance credibility. Additionally, ensure that the units (e.g., Mt of CO2) are explained for clarity.
RESPONSE
The presenting data was taken from Alexandra Troi's article. As she explained in her article, she used statistics data coming from “Bulletin of Housing Statistics for Europe and North America 2004”. This bulletin is published regularly by the United Nations Economic Commission for Europe (UNECE). We added this reference in a footnote (page 3, line 92)
Comment 11: Clarify the term "buffer spaces" in the context of thermal and hygrothermal behavior.
RESPONSE
Agree. The term “buffer spaces” is used in the text to describe internal spaces such as corridor, hall or attic which are integrated in the building’s spatial organization, are not or little heated and which are used specifically as a thermal buffer between the heated spaces and the outside.
The sentence (page 4, second paragraph, line 141) has been modified.
Comment 12: Elaborate on the challenges or considerations associated with generalizing renovation solutions from one case to another, especially regarding the preservation, structural composition, climatic conditions, and occupant behavior.
RESPONSE
Agree. The challenges have been described in a few sentences integrated into the text (page 5, line 193).
Comment 13: Ensure consistency in the use of terms such as "renovation" and "refurbishment." Use these terms consistently throughout the text to avoid confusion.
RESPONSE
In the manuscript, we use mostly the term «renovation» to emphasize the overall building improvement in terms of energy and comfort performance (renovation = upgrading).
In two places in the text (lines 306 and 345), we left the term «refurbishment». This term refers to practices usually used in historic buildings. The objectives are to make improvements to preserve the existing building.
Comment 14: In the section discussing the survey and diagnosis stage, the authors mention focusing on the "four dimensions of the project." It would be helpful to state what these dimensions are, providing clarity explicitly.
RESPONSE
These four dimensions (energy performance, comfort, heritage value and global environmental performance) have been previously defined and discussed in various sections of the article, particularly in the section1 “Introduction”, in point 2.2.1 (page 8, line 332), point 2.3.1 (pages 9/10) and point 2.3.3 (page 10, lines 435)
Comment 15: While it's mentioned that the User Group includes architects, renovation companies, energy efficiency experts, and heritage refurbishment experts, authors might elaborate on how their input influenced decision-making. How did their expertise contribute to the project?
RESPONSE
In section “Methodology”, point 2.2.1 we added sentences explaining the input of the User-Group in the decision-making process and the development of the tools (page 8, line 346). In section “Results and discussion”, point 3.1, we also added a small text to explain the contribution of the User Group (page 20, line 719)
Comment 16: In the Multidisciplinary Approach section, authors mention criteria such as the potential occupancy of the building, comfort improvement, conservation of heritage value, financial cost, and overall environmental impact. Consider briefly specifying how these criteria were weighted or prioritized in decision-making.
RESPONSE
Agree. We added this sentence in point 2.2.1 (page 8, line 368) : “These various criteria were not hierarchically prioritized or weighted. The objective was to evaluate the impact of enhancing energy performance and comfort on the other dimensions of the building, whether positively or negatively.”
Comment 17: When discussing the analysis of heritage significance, consider providing a brief explanation of the four criteria (authenticity, integrity, rarity, and representativeness) and how they were applied.
RESPONSE
Agree. We have included a footnote with a brief explanation of the four criteria and their application (page12, line 466).
Comment 18: Improve references with the following: I.J., Gil Crespo et al., “Climatic analysis methodology of vernacular architecture”, in Vernacular Architecture: Towards a Sustainable Future, pp. 327–32, Taylor & Francis Group, London, 2015; F. Ribera, P. Cucco, I.J.G. Crespo, Energy efficiency features in italian and spanish traditional dwellings, SUSTAINABLE MEDITERRANEAN CONSTRUCTION, 12/2020, pp. 178-183; J. Fernandes et al., The influence of the Mediterranean climate on vernacular architecture: a comparative analysis between the vernacular responsive architecture of southern Portugal and north of Egypt, in Proceedings of the World Sustainable buildings SB14, Barcelona, Spain, 28-30 October 2014; P. Cucco, Dalla Conservazione Integrata di Amsterdam (1975) all’Integrated Approach to Cultural Heritage (2020). Nuove prospettive nello scenario di cambiamenti globali, EDA. ESEMPI DI ARCHITETTURA, 2/2020, pp. 1-10.
RESPONSE
Thank you for these nice references. We have added two of them in the "references" section.

Round 2
Reviewer 1 Report
Comments and Suggestions for Authors
All the requests were satisfactorily responded